# Comparative In Vitro Dissolution Assessment of Calcined and Uncalcined Hydroxyapatite Using Differences in Bioresorbability and Biomineralization

**DOI:** 10.3390/ijms25010621

**Published:** 2024-01-03

**Authors:** Woo Young Jang, Jae Chul Pyun, Jeong Ho Chang

**Affiliations:** 1Korea Institute of Ceramic Engineering and Technology, Jinju 28160, Republic of Korea; 2Department of Materials Science & Engineering, Yonsei University, Seoul 03722, Republic of Korea

**Keywords:** uncalcined hydroxyapatite, bioresorption, biomineralization, calcium citrate, dissolution

## Abstract

This study reports the effect of the not-calcining process on the bioresorption and biomineralization of hydroxyapatite through in vitro dissolution assessment. The prepared calcined hydroxyapatite (c-HAp) and uncalcined hydroxyapatite (unc-HAp) have a particle size of 2 μm and 13 μm, surface areas of 4.47 m^2^/g and 108.08 m^2^/g, and a Ca/P ratio of 1.66 and 1.52, respectively. In vitro dissolution assessments of c-HAp and unc-HAp were performed for 20 days at 37 °C in a citric acid buffer according to ISO 10993-14. During the dissolution, the c-HAp and unc-HAp confirmed an increase in weight, and the calcium and phosphorous ions were rapidly released. The calcium ions released from c-HAp formed rod-shaped particles with a longer and thinner morphology, while in unc-HAp, they appeared thicker and shorter. In the ICP-OES results, the concentrations of calcium elements were initially increased and then decreased by this formation. The rod-shaped particles identified as calcium citrate (Ca-citrate) through the XRD pattern. The calcium content of Ca-citrate particles from unc-HAp was higher than that from c-HAp. The unc-HAp demonstrated non-toxic properties in a cytotoxicity evaluation. Therefore, due to its higher bioresorption and biomineralization, unc-HAp exhibits enhanced biocompatibility compared to c-HAp.

## 1. Introduction

Hydroxyapatite is a ceramic material that exhibits certain similarities to human bone and has been utilized for a very long time as an artificial bone regeneration and replacement material due to its excellent physicochemical and biocompatibility [1,2]. Hydroxyapatite can be synthesized through a variety of methods, including dry, wet, and high-temperature pyrolysis processes. Most commercially available hydroxyapatites consist of calcined hydroxyapatite (c-HAp), which is produced through a high-temperature pyrolysis process [3,4,5,6,7,8,9,10,11,12,13,14]. Produced through a high-temperature pyrolysis process, c-HAp is manufactured by applying only friction and thermal energy to powdered precursor chemicals (calcium and phosphate) [3,4,5,9,10,11,12,13,14]. Many impurities are generated during this process, making it difficult to precisely control the composition and morphology. The composition, morphology, and particle size can be easily controlled by adjusting the temperature, pressure, pH, reaction time, etc. [6,7,8]. In addition, external materials such as chitosan, polymeric binders, and titanium oxide are often added to control the shape of the particles [15,16,17].

Synthetic hydroxyapatite is mainly used for bone defects, but it has the disadvantage of very low bone regeneration efficiency compared to autologous bone. However, autologous bone is limited in quantity and has the potential for secondary damage, pain, and infection. Therefore, research on biomaterials for bone regeneration is continuously reported [18,19,20]. In particular, the development of bone materials such as organic/hydroxyapatite and polymer/hydroxyapatite composites is aimed at maximizing bone regeneration efficiency [21,22,23,24,25,26,27,28]. Although hydroxyapatite is characterized by bone conduction, bone regeneration, and biocompatibility, its very low solubility results in little body absorption. To address these issues, various studies using uncalcined hydroxyapatite (unc-HAp) have been reported. For example, bone screws and plates combining biodegradable polylactic acid and unc-HAp show improved biodegradability and osteoinductive properties and have been commercialized in China and Japan [29,30,31].

The main advantage of unc-HAp compared with c-HAp is that it does not require a high-temperature calcination process. In biomaterials, unc-HAp has the advantages of a higher specific surface area and faster elution of calcium and phosphate, resulting in a better biomineralization effect than c-HAp, which has undergone high-temperature calcination. These properties are the most important differences between c-HAp and unc-HAp, but studies that mention them are not reported, or if they are, they are simply comparing them based on XRD patterns and FT-IR results [32]. Also, the reason for choosing unc-HAp rather than c-HAp is its solubility, which has been applied as a material in polymer/hydroxyapatite composites [25,29,33,34,35,36,37].

In particular, the currently commercialized Osteotrans™ from Takiron, Japan, incorporates unc-HAp and is slowly absorbed over a period of 4–6 years while binding to bone in the body, making it an excellent material for orthopedic and craniofacial applications. However, the unc-HAp used in these polymer/hydroxyapatite composites has not been individually characterized, despite being the main material. In addition, the structural changes and kinetic evaluation of unc-HAp over time have also not been reported. Although the biological behavior of unc-HAp is important because it must interact with bone, it has been shown that dissolution behavior such as ion release, precipitation, and ionic changes occur first. Therefore, it is important to understand the properties of unc-HAp from a material perspective.

In order for synthesized unc-HAp to have biological and medical applications, a number of conditions must be satisfied. The most important of these conditions is biocompatibility, which refers to the ability of a biomaterial to perform its intended function in the body without causing adverse effects. It is also important to ensure that the biomaterial does not elicit harmful reactions over time, as it will be applied in an environment within the body. To confirm this, in vitro dissolution assessments are performed in various environments such as phosphate buffered saline [38,39] and simulated body fluid [40,41,42,43,44,45,46], which is an essential element for the safety of biomaterials [47,48]. In addition, since bones constantly undergo mineralization and remodeling processes, it is necessary to observe the bioresorption and biomineralization that occur during the dissolution of biomaterials to confirm their role as synthetic bones. Bioresorption refers to the process of first removing minerals from biomaterials. Biomineralization, on the other hand, is the process of producing crystalline minerals, which allows the formation of biominerals such as bones and teeth. Evaluating these two phenomena can determine the long-term safety, compatibility, and various possibilities for the bone tissue regeneration of biomaterials. However, in vitro dissolution evaluation in neutral solutions such as phosphate-buffered saline and simulated body fluids (SBF) cannot fully confirm the bioresorption and biomineralization of unc-HAp in a short period of time, because the elution of calcium and phosphorus ions from unc-HAp is too slow in neutral solutions. Therefore, ISO 10993-14 (Biological evaluation of medical devices for ceramic) also suggests using citric acid solution [49], which has a fast elution of calcium and phosphorus ions, in addition to a neutral Tris-buffer solution. The use of citric acid, a weak acid, is suggested because in solutions that are too strongly acidic, the elution of calcium and phosphorus ions occurs very rapidly, but leads to poor biomineralization formation. Therefore, citric acid according to ISO 10993-14 was also used in this study. The bioresorption and biomineralization of hydroxyapatite were compared by in vitro dissolution assessments to determine the effect of not-calcining process. The physicochemical properties of c-HAp and unc-HAp were measured by particle size, crystal structure, elemental composition, and compressive strength. In addition, the in vitro dissolution assessment of c-HAp and unc-HAp was evaluated in citric acid according to ISO 10993-14 [49]. Newly formed particles were confirmed during the dissolution process of c-HAp and unc-HAp, which were rod-shaped but differed in thickness and length. These particles were formed based on the differences in the bioresorption and biomineralization of c-HAp and unc-HAp, and a mechanism for this was proposed. Furthermore, cell viability showed that unc-HAp was less toxic than c-HAp.

## 2. Results and Discussion

Figure 1 shows the TEM image, particle size, and XRD pattern of c-HAp and unc-HAp. Figure 1a shows TEM images of c-HAp and unc-HAp. The particle shape of c-HAp was observed to be polygonal, whereas that of unc-HAp was spherical. This change occurred as the thermal energy received during calcination increased the density of the particles due to energy exchange. Thus, the individual particle size of c-HAp averaged 183 nm, which was larger than unc-HAp (average 20 nm). The shape and size of these hydroxyapatite particles can be determined through various synthesis methods [6,7,8]. In particular, the shape and average size can be easily controlled using wet synthesis. In addition, the desired properties can be obtained by changing conditions such as the starting materials, acid/base addition rate, pH, and reaction temperature [50,51].

Figure 1b shows particle size distribution through PSA and SEM images of c-HAp and unc-HAp. The particle sizes on the d_50_ values of c-HAp and unc-HAp were 2 μm and 13 μm, respectively. The shapes observed in the SEM images were all polygonal-shapes, which were clusters of nanometer particles. The size of the particles was the same as the particle size distribution result, in the order of c-HAp < unc-HAp. This is attributed to the thermal energy causing contraction between the particles during the calcination process, ultimately resulting in a smaller particle size. Therefore, the average particle size of c-HAp is seven times smaller than that of the unc-HAp.

Figure 1c shows the XRD patterns of c-HAp and unc-HAp. Both c-HAp and unc-HAp obtained the characteristic peaks at 25.8°, 28.9°, 31.8°, 32.2°, 32.9°, 34.1°, 39.8°, 46.7°, 49.5°, and 53.2°. Among these peaks, the (002), (120), (121), (112), (300), (202), (130), (222), (213), and (004) plane of hydroxyapatite [Ca_10_(PO_4_)_6_(OH)_2_, JCPDS#01-080-6199] with the highest intensity was confirmed at 2θ = 31.8°. Furthermore, the c-HAp exhibited an overall sharpness, while the unc-HAp showed a broader profile. These results are attributed to the influence of the calcination of hydroxyapatite, and the XRD patterns become sharper as the calcined temperature increases [52,53].

Figure 2 shows the nitrogen adsorption–desorption isotherms, BET parameters, Ca/P ratio, cell viability, and compressive and fracture strength of c-HAp and unc-HAp. Figure 2a shows the nitrogen adsorption–desorption isotherms of c-HAp and unc-HAp. Both c-HAp and unc-HAp are the most consistent with type IV among the physisorption isotherms provided by IUPAC. Also, the surface area was larger for unc-HAp compared to c-HAp. This indicates a decrease in the surface area as calcination progresses, with the hysteresis loop changed to an H1 type from an H5 type [54,55].

Figure 2b shows the surface area, pore volume, and pore size of c-HAp and unc-HAp according to BET analysis. The surface areas of c-HAp and unc-HAp were 4.47 m^2^/g and 99.35 m^2^/g, respectively. The pore volumes and pore sizes were 0.02 cm^3^/g and 0.60 cm^3^/g, and 20.89 nm and 25.32 nm, corresponding to c-HAp and unc-HAp, respectively. The surface area, pore volume, and pore size of c-HAp were much lower than those of the unc-HAp. This phenomenon occurs as calcination progresses, leading to a denser internal structure and the occlusion of some pores. Additionally, under the influence of thermal energy, the particles bond together and the pore size becomes smaller in this process.

The Ca/P molar ratio in calcium phosphate-based biomaterials is highly significant as it determines not only the biomaterials’ chemical composition, but also impacts biocompatibility and mechanical properties. Each calcium and phosphorous elements contents in c-HAp and unc-HAp were determined using an ICP-OES analysis, and their Ca/P molar ratio was calculated as shown in Figure 2c. The calcium elements contents were 42.51 wt.% and 42.03 wt.% for c-HAp and unc-HAp, respectively, while the phosphorous elements contents were 19.85 wt.% and 21.36 wt.%, respectively. The Ca/P molar ratios were 1.66 for c-HAp and 1.52 for unc-HAp. In general, the Ca/P molar ratio of c-HAp, as an implantable biomaterial, is known to be 1.4 to 1.7 [56,57,58,59].

The cell viability of both c-HAp and unc-HAp was demonstrated for 24 h using RAW 264.7 cells, as shown in Figure 2d. The cultured RAW 264.7 cells were dispensed and incubated at 37 °C, 5% CO_2_. After 24 h, the cells were treated with c-HAp and unc-HAp at concentrations of 25, 100, 250, 500, and 1000 μg/mL. The cell viability of c-HAp and unc-HAp at 1000 μg/mL exhibited 93% and 92%, respectively. Therefore, both c-HAp and unc-HAp have been demonstrated to be suitable for use as biomaterials.

Figure 2e shows the mechanical properties through the compressive strength–strain curve and fracture strength of c-HAp and unc-HAp. To obtain compression test results, pellets of c-HAp and unc-HAp were manufactured using a press molding with a circular die 20 mm in diameter for 2 g of powder, respectively. The compressive strength of c-HAp was higher than that of unc-HAp, while the strain was lower. Additionally, the fracture strength was determined to be 54.2 MPa for c-HAp and 43.1 MPa for unc-HAp. These results are due to the fact that the particle size of c-HAp is about seven times smaller than that of the unc-HAp, allowing it to be manufactured into denser pellets. Also, due to its higher density, the c-HAp exhibits greater compressive and fracture strength compared to the unc-HAp, resulting in easier deformation. This indicates that the mechanical properties of c-HAp and unc-HAp are suitable for use as biomaterials.

Figure 3 shows the weight change, concentrations of calcium and phosphorus ions, XRD patterns, dissolution efficiency, dissolution rate, and SEM images of c-HAp and unc-HAp, which were evaluated to in vitro dissolution according to ISO 10993-14 [49]. The weight changes of c-HAp and unc-HAp drastically decreased to 12% and 24%, respectively, compared to the initial weight for 1 day, as shown in Figure 3a. The weight loss occurred in c-HAp and unc-HAp due to the removal of mineral constituents. The dissolution of calcium phosphate bioceramics is primarily related to their chemical composition and crystalline characteristics. Also, it is influenced by factors such as the surface area and particle size [60,61,62]. Increasing the surface area enhances the area in contact with the fluids, which leads to a faster dissolution rate. Also, spherical calcium phosphate particles tend to leave small and deep pores, a phenomenon attributed to the incomplete calcination of ceramics when lower calcination temperatures and times are employed [63]. Therefore, due to the higher surface area and pore volume of unc-HAp compared to c-HAp, a more significant weight loss due to dissolution was observed. Subsequently, the c-HAp weight change increased for 16 days, while the unc-HAp weight change increased for 8 days. The weight change after 20 days of dissolution, compared to the initial weight, was increased by 3% for c-HAp and decreased by 7% for unc-HAp. In general, the weight of the powder decreases according to the dissolution time. However, the weight of c-HAp and unc-HAp showed that the weight of the c-HAp and unc-HAp increased continuously after 1 day. This indicates the conclusion that both c-HAp and unc-HAp generate unknown materials during the dissolution process.

To confirm this, the changes in the concentrations of the calcium and phosphorus ions released from the c-HAp and unc-HAp dissolution were confirmed by ICP-OES analysis, as shown in Figure 3b. The concentrations of calcium elements decreased for both c-HAp and unc-HAp after an initial increase, while the concentrations of phosphorus elements exhibited an increasing trend. Before 1 day to dissolution (5, 15, 30, 45 min and 2, 8, 14, 18 h), the highest calcium elements concentrations for c-HAp and unc-HAp were observed at 39,780 mg/L after 18 h and 34,420 mg/L after 14 h, respectively. Subsequently, the concentrations of calcium elements decreased. Finally, the concentrations of calcium and phosphorus elements at 20 days were 14,423 mg/L and 38,796 mg/L, and 14,260 mg/L and 50,757 mg/L corresponding to c-HAp, and unc-HAp, respectively. Consequently, the concentrations of the calcium ions released from both c-HAp and unc-HAp decreased as the dissolution time increased, but the concentrations of the phosphorus ions released from unc-HAp increased rather than that of c-HAp. These results confirm the generation of substances containing calcium in both c-HAp and unc-HAp. Furthermore, due to the higher concentrations of calcium ions in unc-HAp compared to c-HAp, the dissolution of unc-HAp is more effective.

Figure 3c shows the XRD patterns and crystallinity of c-HAp and unc-HAp after 20 days. The patterns of both c-HAp and unc-HAp revealed hydroxyapatite, albeit with some distinct characteristic peaks. The characteristic peaks of c-HAp were observed at 21.8°, 52.1°, 53.2°, 55.9°, and 57.1°, corresponding to the (200), (402), (004), (322), and (313) planes of hydroxyapatite [Ca_5_(PO_4_)_3_OH, JCPDS#01-076-8436]. On the other hand, characteristic peaks of unc-HAp exhibited at 52.0° and 53.1°, corresponding to the (303) and (411) planes of hydroxyapatite [Ca_4.758_(H_0.21_(PO_4_)_3_)(OH)_0.726_, JCPDS#01-074-9764]. These differences are a result of alterations in hydroxyapatite (JCPDS#01-080-6199) due to dissolution. These results indicate respective changes in hydroxyapatite (JCPDS#01-080-6199) due to the process of dissolution. The crystallinity of c-HAp and unc-HAp after 0 and 20 days was calculated based on the following equation [64,65].
$$\mathrm{Crystallinity}\;(\%)=\frac{\mathrm{Area}\;\mathrm{of}\;\mathrm{crystalline}\;\mathrm{peaks}}{\mathrm{Area}\;\mathrm{of}\;\mathrm{all}\;\mathrm{peaks}}\;\times 100$$

The crystallinity of c-HAp and unc-HAp on day 0 was calculated to be 80.1% and 77.4%, respectively. On day 20, these values decreased to 65.3% and 54.4%, respectively. The crystallinity generally depends on the calcination temperature [66,67]. A calcination process involving high thermal energy substantially affects the physical properties, thus causing phase changes and microstructure changes in the particles. Due to these changes, the crystallinity is affected by the bonding and strong aggregation between each particle. Therefore, the crystallinity of c-HAp on day 0 was higher than that of unc-HAp. However, the crystallinity of c-HAp and unc-HAp after 20 days decreased by 14.8% and 23.0%, respectively. This is due to the release of the mineral constituents composed of calcium cations and phosphate anions during the dissolution process. Therefore, unc-HAp demonstrates better dissolution compared to c-HAp.

Figure 3d shows the concentration of calcium and phosphorus ions released over time as dissolution efficiency. The dissolution efficiency of c-HAp and unc-HAp was calculated based on the following equation.
$$\mathrm{Dissoultion}\;\mathrm{efficiency}\;\left(\%\right)=\frac{\mathrm{released}\;\mathrm{concentration}\;\mathrm{of}\;\mathrm{ion}}{\left(\mathrm{initial}\;\mathrm{weight}\;\mathrm{of}\;\mathrm{HAp}\times \mathrm{atomic}\;\mathrm{weight}\right)}\times 100$$

The dissolution efficiency of calcium exhibited a rapid increase for both c-HAp and unc-HAp, remaining constant up to 14 h. The dissolution efficiency of c-HAp and unc-HAp at 14 h was 8.0% and 8.2%, respectively. In contrast, the dissolution efficiency of phosphorus was higher for unc-HAp compared to c-HAp. The dissolution efficiency of c-HAp and unc-HAp at 20 days was 20.6% and 27.8%, maintaining this efficiency from 10 days onwards, respectively. This means that the dissolution rate of unc-HAp is faster than that of c-HAp. To support these findings, the dissolution rates of c-HAp and unc-HAp are shown in Figure 3e. The dissolution rate was calculated using the following formula and plotted as a linear relationship.
$$\mathrm{Disso}l\mathrm{ution}\;\mathrm{rate}\;(\mathrm{ppm}/\mathrm{time})=\frac{∆\left[\mathrm{released}\;\mathrm{concentration}\;\mathrm{of}\;\mathrm{ion}\right]}{∆[\mathrm{time}]}$$

The dissolution rate of calcium is higher for unc-HAp compared to c-HAp due to the higher slope. Also, the dissolution rates of c-HAp and unc-HAp became equal at 4 h. Subsequently, as the dissolution time increased, the dissolution rates decreased. The dissolution rates of c-HAp and unc-HAp reached 0 ppm/h at 14 h and 10.5 h, respectively. The dissolution rate of phosphorus showed a higher slope for unc-HAp compared to c-HAp. The dissolution rates of c-HAp and unc-HAp became equal on day 6 and continued to decrease. Furthermore, the dissolution rates of c-HAp and unc-HAp reached 0 ppm/days on day 11 and day 8, respectively. These results demonstrate that the dissolution rate of unc-HAp is faster than that of c-HAp.

Figure 3f shows the morphological changes in c-HAp and unc-HAp observed over the course of dissolution. Both c-HAp and unc-HAp confirmed the formation of new rod-shaped particles after dissolution. The rod-shaped particles were more abundant in unc-HAp compared to c-HAp, and their quantity increased with time. Additionally, the rod-shaped particles in unc-HAp were thicker and shorter than those in c-HAp. These results could be attributed to the decrease in the concentration of calcium ion and dissolution rate. In particular, it is expected that the thicker and shorter rod-shaped particles in unc-HAp are associated with the rapid dissolution rate. Thus, released calcium was used in the formation of new particles, and as their quantity increases, they tend to become thicker and shorter. These particles are likely to be calcium citrate (Ca-citrate) since they are formed as a result of the removal of calcium ions in a citric acid solution [68]. Consequently, this phenomenon is associated with bioresorption and biomineralization. In general, bioresorption involves resorption cells breaking down the biomaterials and releasing the ions, resulting in a transfer of calcium and phosphate from the biomaterials to the blood. Subsequently, the emitted ions are employed in bone remodeling, a process encompassing biomineralization. In other words, both c-HAp and unc-HAp are dissolved by the citric acid buffer, resulting in the formation of rod-shaped particles through the release of ions. Therefore, it can be inferred that there is a relationship between these processes and bioresorption and biomineralization.

To confirm the characterization of newly formed rod-shaped particles, FIB-SEM, EDS, and XRD analyses were used, as shown in Figure 4. Figure 4a shows an FIB-SEM (Focused ion beam-scanning electron microscopy) image, aspect ratio, and EDS (Energy dispersive spectrometer) analysis of a cross-section by focused ion beam using an acceleration voltage of 2 kV and a current of 50 pA. The confirmed rod-shaped particles of c-HAp and unc-HAp were either longer or thinner, and shorter or thicker, respectively. These were subjected to carbon deposition followed by milling using FIB, and their images were observed using SEM. Both c-HAp and unc-HAp exhibited the carbon layer, with the cross-sectional size of rod-shaped particles being larger in unc-HAp. The thickness and width of these were 0.45 μm and 1.32 μm in c-HAp, and 0.69 μm and 1.82 μm in unc-HAp. Consequently, the aspect ratio of c-HAp and unc-HAp was determined to be 659:227 and 1823:687, respectively. Also, EDS analysis was used to identify the constituent components of the rod-shaped particles. The results indicated the calcium, carbon, and oxygen with, respectively, 4.9 wt.%, 81.1 wt.%, and 14 wt.% in c-HAp, and 11.3 wt.%, 71.1 wt.%, and 17.6 wt.% in unc-HAp. Ultimately, unc-HAp exhibited a three-time higher aspect ratio and calcium content than c-HAp. This suggests a correlation between the calcium content and particle size.

Figure 4b shows the X-ray diffraction (XRD) pattern and molecular modeling of c-HAp and unc-HAp after 0 and 20 days of dissolution. The XRD pattern after 20 days of c-HAp and unc-HAp was similar to the pattern of 0 day, but the appearance of a new peak confirmed the phase change. In c-HAp, new peaks were observed at 5.7°, 10.4°, 11.5°, 17.0°, 18.7°, 22.6°, 28.7°, and 28.9° and were confirmed to correspond to the (200), (201), (400), (600), (−601), (−112), (403), and (1000) planes of [Ca_3_(C_6_H_5_O_7_)_2_∙4H_2_O, JCPDS#00-028-2003]. In unc-HAp, new peaks were observed at 5.7°, 9.1°, 11.4°, 12.1°, 17.1°, 17.3°, 18.2°, 23.1°, 27.1°, 28.7°, and 28.8° and were confirmed to correspond to the (001), (010), (002), (0–11), (003), (−110), (112), (121), (0–14), (−1–13), and (0–23) planes of [Ca_3_(C_6_H_5_O_7_)_2_∙(H_2_O)_4_, JCPDS#01-084-5956]. These new peaks demonstrated changes resulting from dissolution. Additionally, both c-HAp and unc-HAp were confirmed to be Ca-citrate by phase analysis, while they exhibited different JCPDS# and crystal systems. In particular, the crystal system of c-HAp and unc-HAp were identified as monoclinic (Ca-citrate-1) and triclinic (Ca-citrate-2), respectively. Monoclinic crystals have three unequal axes, which are inclined to one another, with the third axis being perpendicular. On the other hand, triclinic crystals have three unequal axes with oblique angles. The difference between these two crystal systems is in their symmetry. The triclinic crystal system does not possess any symmetry elements because all its axes are unequal. Therefore, Ca-citrate-2 is more likely to have an irregular atomic arrangement compared to Ca-citrate-1. Based on these predictions, the molecular modeling of Ca-citrate-1 and Ca-citrate-2 was depicted. The Ca-citrate consists of repeating units connected by calcium ions. Therefore, two di-citrate molecules can bind with three calcium ions [69,70,71,72]. In more detail, citrate binds with calcium ions through chelation, forming a complex that results in the creation of a Ca-citrate complex. This structure exhibits crystalline characteristics, manifesting an elongated form and a consistent molecular arrange-ment due to the binding of two citrate molecules with three calcium ions. In addition, the calcium in Ca-citrate can form a maximum of six bindings, including coordination bonds, thus enabling growth in a stacking form [65,66]. As a result, Ca-citrate is composed of repetitive units that exhibit an increase in length and a regular molecular arrangement, thereby demonstrating the characteristics of a crystal. This chemical structure is similar to that of Ca-citrate-1. By contrast, the thickness and length of the rod-shaped particles in Ca-citrate-2 were the opposite. Because Ca-citrate-2 has three times higher calcium content compared to Ca-citrate-1. The presence of excessive released calcium ions by a higher dissolution rate constantly combines and condenses with numerous citrate ions, leading to excessive nucleation. As a result, the remaining regions of Ca-citrate bind randomly, and during this process, the calcium ions are rapidly depleted, leading to an increase in thickness and a decrease in length. Therefore, the rod-shaped particles of Ca-citrate-1 and Ca-citrate-2 have different sizes, with d_1_ < d_2_.

Figure 4c shows the schematic representations on the formation of the rod-shaped particles from c-HAp and unc-HAp in citric acid. The calcium and phosphate ions are released from c-HAp and unc-HAp due to citric acid. The released calcium ions combine with di-citrate ions in the solution to form Ca-citrate. In this process, since more calcium ions are released from unc-HAp compared to c-HAp, more Ca-citrate is formed at the same time. Despite this, the remaining calcium ions continue to contribute to the formation of Ca-citrate in unc-HAp, leading to the growth of Ca-citrate-2. On the other hand, its growth into Ca-citrate-1 can be attributed to the limited supply of calcium ions in c-HAp.

Figure 5a shows the cell viability of c-HAp and unc-HAp on the 20 days of dissolution using RAW 264.7 cells. The cultured RAW 264.7 cells were treated with concentrations of 25, 100, 250, 500, and 1000 μg/mL of c-HAp and unc-HAp and incubated. The results showed that the cell viability of eluted c-HAp for 20 days was 78% for 25 μg/mL, and that as the concentration increased, the cell viability decreased. However, unc-HAp eluted for 20 days showed more than 80% cell viability, up to a concentration of 500 μg/mL. Therefore, the Ca-citrate-1 have toxicity and affect the decrease in the cell viability of c-HAp on the 20 days of dissolution. Moreover, Ca-citrate-2 showed almost no effect on cell viability, which was attributed to the high calcium content.

These results were confirmed from the calcium/carbon contents of Ca-citrate-1 and Ca-citrate-2, as shown in Figure 5b. The calcium/carbon contents were calculated based on the results of EDS analysis. The calcium/carbon contents of Ca-citrate-1 and Ca-citrate-2 were 6% and 16%, respectively. Ca-citrate-2 exhibited three times higher calcium content than Ca-citrate-1. The calcium content is a factor that can influence cell viability. The released calcium ions increase the local ion concentration, which has a significant impact on cell survival and proliferation, as well as the process of osteogenesis [73,74,75]. Therefore, unc-HAp should be considered a biocompatible material in the human body, and it exhibits better biological activity than c-HAp.

Figure 6 shows the bioresorption and biomineralization in c-HAp and unc-HAp during the dissolution in citric acid. When c-HAp and unc-HAp are immersed in a citric solution, dissolution begins and calcium ions are gradually released against the dissolution time. This process corresponds to bioresorption as c-HAp and unc-HAp are dissolved and release ions into the solution. The unc-HAp exhibits a higher of bioresorption compared to c-HAp. However, the concentration of calcium ions decreases drastically after a certain period, after which it remains constant. As the calcium ions decrease, the formation of Ca-citrate occurs, a process associated with biomineralization. The formation of Ca-citrate was observed through SEM analysis. Biomineralization was higher in unc-HAp compared to c-HAp. This is attributed to the higher concentration of calcium ions, which leads to excessive Ca-citrate binding. As a result, Ca-citrate-2 forms in unc-HAp, characterized by a shorter and thicker appearance. In contrast, due to the lower calcium ion concentration, c-HAp forms Ca-citrate-1, resulting in a longer and thinner appearance. Therefore, this concentration of calcium ion changes the result in the bioresorption and biomineralization in c-HAp and unc-HAp.

## 3. Materials and Methods

### 3.1. Materials

Calcium hydroxide, ammonium dihydrogen phosphate, Calcium nitrate tetrahydrate, citric acid, and sodium hydroxide were all purchased from Sigma-Aldrich (Korea). Diammonium hydrogen phosphate was purchased from Junsei (Japan). Ammonium hydroxide and hydrochloric acid were purchased from Daejung (Korea).

### 3.2. Preparation of Uncalcined Hydroxyapatite (Unc-HAp) and Calcined Hydroxyapatite (c-HAp)

Uncalcined hydroxyapatite was prepared by the following. Calcium nitrate tetrahydrate (6.093 kg) and diammonium hydrogen phosphate (2.641 kg) were each dissolved in deionized water (40 L). The diammonium hydrogen phosphate solution was added drop-wise to the calcium nitrate tetrahydrate solution and constantly stirred at room temperature. Then, ammonium hydroxide (10 L) was slowly added drop-wise to increase the pH to 10, and stirring was maintained for 48 h. After stirring, the solution was aged for 24 h, and the obtained unc-HAp powder was washed three to four times with deionized water before being completely dried in an oven at 80 °C. The calcined hydroxyapatite was obtained by uncalcined hydroxyapatite at 1250 °C for 6 h.

### 3.3. Cell Viability

The cell viability of the c-HAp and unc-HAp powder was determined using an MTT assay. For co-culturing with RAW264.7 cells, the c-HAp and unc-HAp powder was autoclaved, and Dulbecco’s Eagle Medium (Hyclone, USA) was added. After sonication for more than 30 min for dispersion, the c-HAp and unc-HAp suspension was incubated. The treated cells were then washed with a medium and incubated with MTT reagent. At this point, the optical density (OD) of each was measured using a microplate reader at 570 nm wavelength. The cell viability of the treated cells was calculated relative to the control (c-HAp and unc-HAp powder 0 μg/mL) using the following formula [76,77].
$$\mathrm{Cell}\;\mathrm{Viability}\;(\%)=\frac{{\mathrm{OD}}_{570}\;\mathrm{of}\;\mathrm{treated}\;\mathrm{cells}}{{\mathrm{OD}}_{570}\;\mathrm{of}\;\mathrm{control}\;\mathrm{cells}}\times 100$$

### 3.4. In Vitro Dissolution Assessments

The dissolution assessments of c-HAp and unc-HAp powder were conducted as follows: To prepare the citric acid solution, citric acid (21 g) was dissolved in deionized water (0.2 L). Then, 1 mol of sodium hydroxide (0.2 L) and deionized water were added until reaching a final volume of 1 L. Finally, 1 mol of hydrochloric acid was added to prepare a citric acid buffer solution of pH 3. Citric acid buffer solution (0.04 L) was then added to each c-HAp (2 g) and unc-HAp powder (2 g) and observed through a shaking incubator at 37 °C and 2 Hz for 20 days. Lastly, the decomposed c-HAp and unc-HAp powder was separated using a centrifuge, washed three times with deionized water, and completely dried in an oven at 80 °C.

### 3.5. Characterizations

The morphology of the c-HAp and unc-HAp powder was determined by transmission electron microscopy (TEM) using a JEM-2100Plus (JEOL, Japan), field emission scanning electron microscopy (SEM) using a MIRA 3 (TESCAN, Czech), and focused ion beam (FIB) using a Helios 5 UC (ThermoScientific, USA) at an acceleration voltage of 200 kV and 2 kV, respectively. The particle sizes of the c-HAp and unc-HAp powder were measured with a particle size analyzer (PSA) using an LA-960 (HORIBA, Japan). The phase of the c-HAp and unc-HAp powder was determined through X-ray diffraction (XRD) in the 2-theta range 5–60° using Miniflex600 (RIGAKU, Japan). Brunauer-Emmett-Teller (BET) analyses were performed using an ASAP 2420 (Micromeritics, USA). The compressive strength of the c-HAp and unc-HAp powder was tested by a universal testing machine (UTM) using a Z050TH (Zwick Roell, Germany) with the crosshead speed of 1.3 mm/min. The compressive and fracture strength was determined by the stress–strain curve obtained from the compression test. The calcium and phosphorus content of the c-HAp and unc-HAp powder and the eluted filtrate were measured by inductively coupled plasma optical emission (ICP-OES) using an Avio500 (Perkin Elmer, USA).

## 4. Conclusions

The synthesized unc-HAp had a smaller primary particle size than c-HAp, but a larger aggregated particle size. In addition, the surface area and pore volume were far higher. With these material properties, the bioresorption and biomineralization of unc-HAp were higher than c-HAp. Accordingly, more newly formed rod-shaped particles were observed in unc-HAp as the dissolution time increased, and the morphology of these particles was shorter and thicker than the newly formed rod-shaped particles in c-HAp. These particles were identified as Ca-citrate in both c-HAp and unc-HAp, but unc-HAp had much higher calcium content and cell viability. Therefore, unc-HAp is more suitable as a main material for bone regeneration than c-HAp. These results ensure the long-term safety and biocompatibility of unc-HAp, indicating its potential application as a key material in the medical industry.

## Figures and Tables

**Figure 1 ijms-25-00621-f001:**
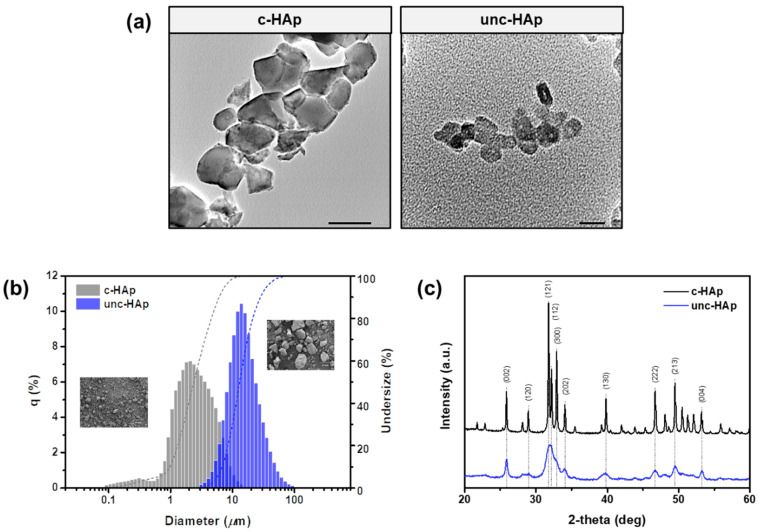
Comparison of c-HAp and unc-HAp characterizations. (**a**) TEM images, (**b**) particle size distribution and SEM image, and (**c**) XRD pattern.

**Figure 2 ijms-25-00621-f002:**
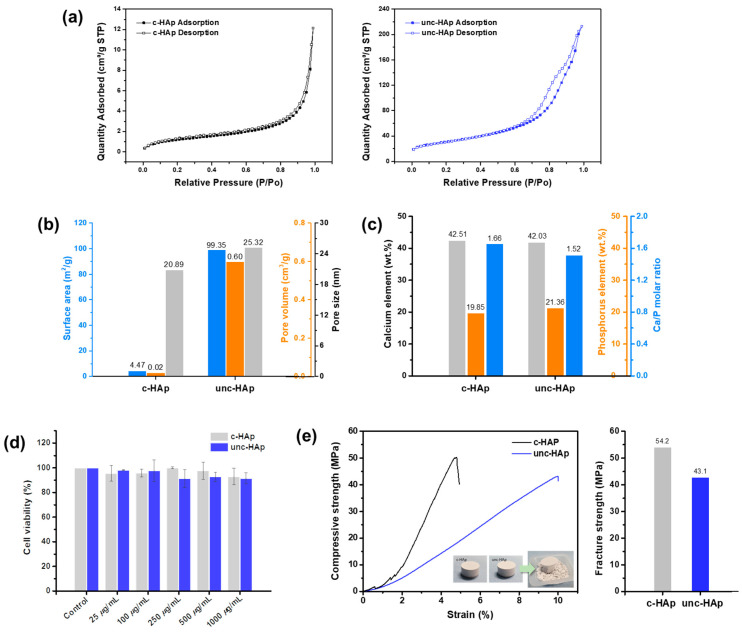
(**a**) Nitrogen adsorption–desorption isotherms, (**b**) BET parameters of surface areas, pore volume, and pore size, (**c**) contents of calcium and phosphorous elements and Ca/P molar ratios, (**d**) cell viability, and (**e**) compressive strength–stain curve and fracture strength of the c-HAp and unc-HAp.

**Figure 3 ijms-25-00621-f003:**
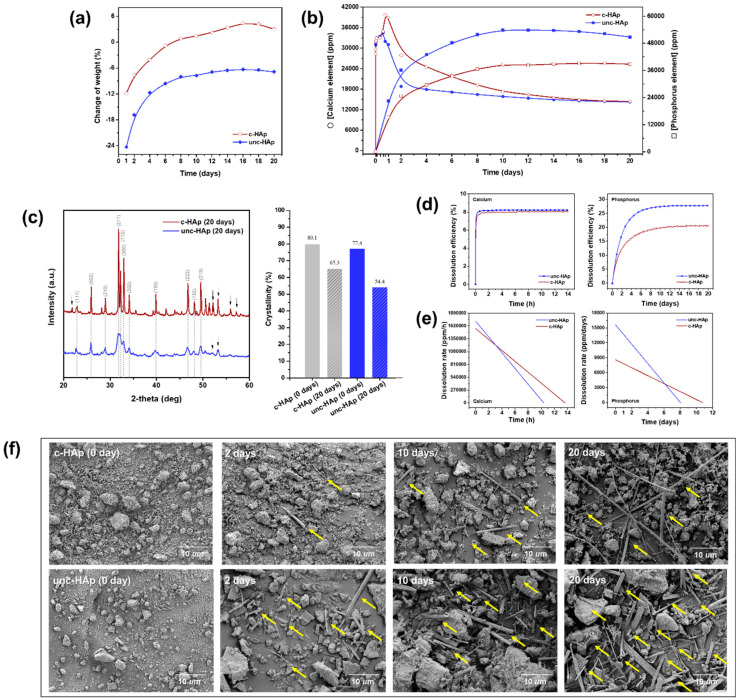
Comparison of in vitro dissolution assessments of c-HAp and unc-HAp. (**a**) Weight of powder, (**b**) calcium and phosphorus elements concentrations in the eluted filtrate by ICP-OES analysis, (**c**) XRD pattern and crystallinity, (**d**) dissolution efficiency, (**e**) dissolution rate, and (**f**) SEM images as a function of the time.

**Figure 4 ijms-25-00621-f004:**
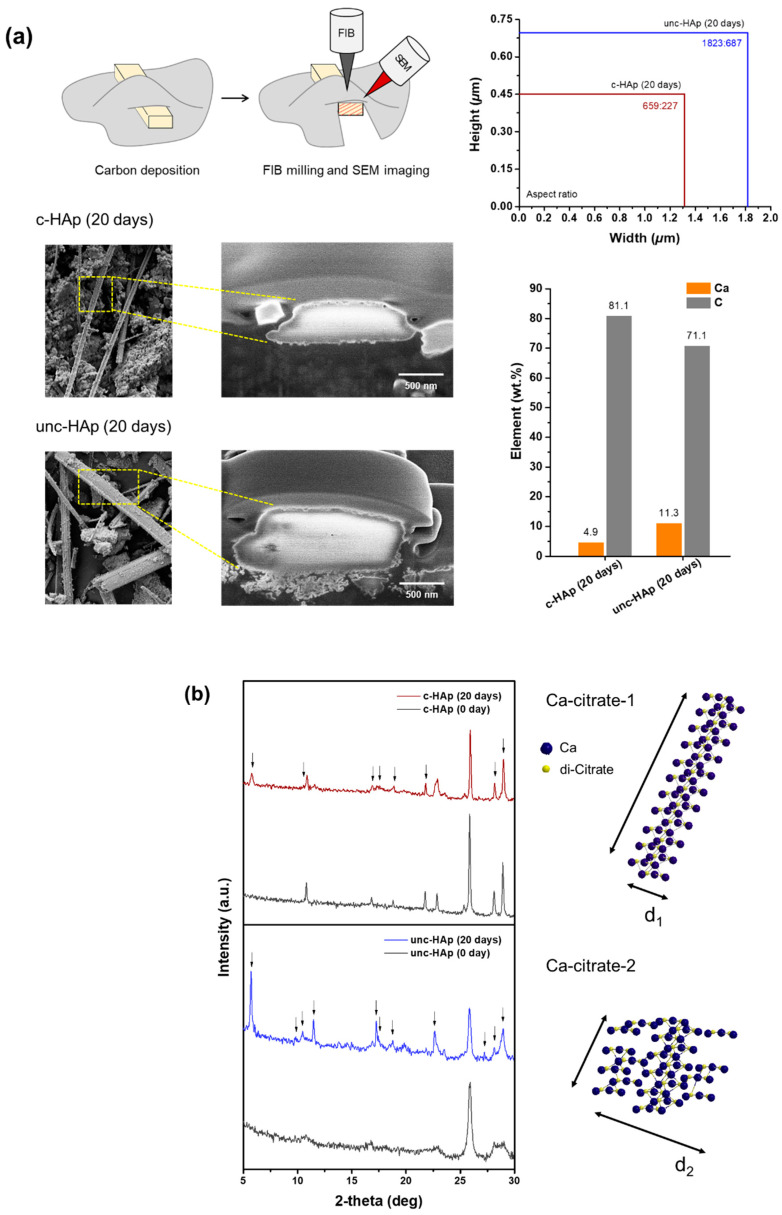
(**a**) FIB-SEM images showing cross-sectional views, aspect ratio, and EDS results of rod-shaped particles, (**b**) XRD pattern and molecular modeling of the c-HAp and unc-HAp at 20 days, and (**c**) schematic representations of c-HAp and unc-HAp in citric acid.

**Figure 5 ijms-25-00621-f005:**
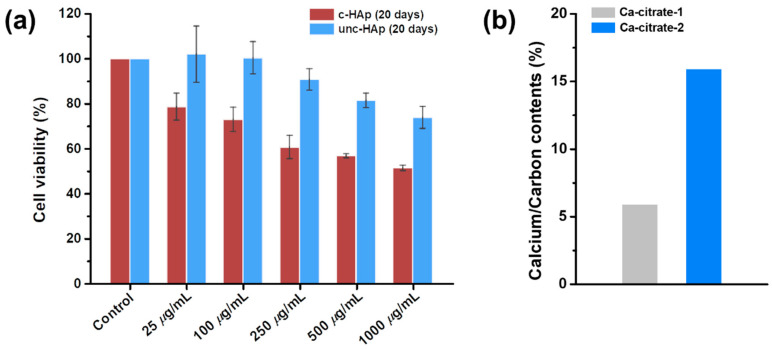
(**a**) Cell viability through an MTT assay observed during in vitro dissolution assessment of c-HAp and unc-HAp at 20 days, and (**b**) calcium/carbon contents of Ca-citrate-1 and Ca-citrate-2.

**Figure 6 ijms-25-00621-f006:**
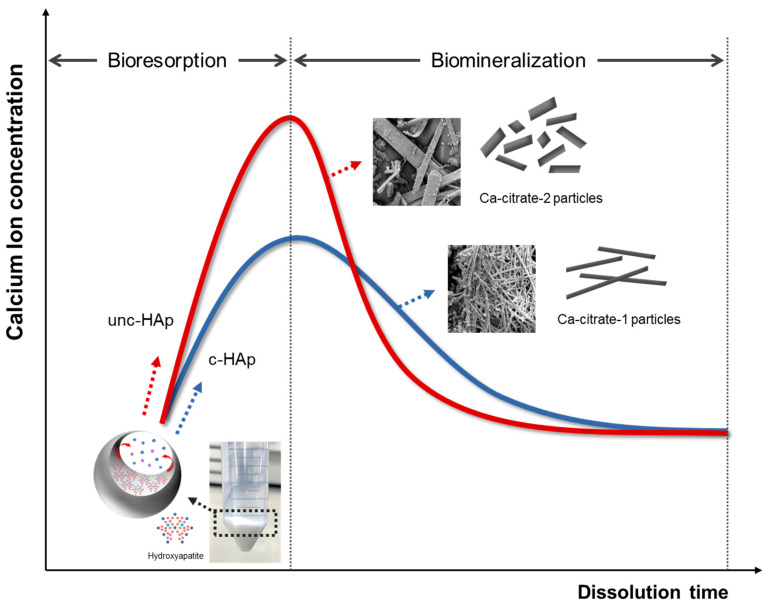
Changes in calcium ion concentration with bioresorption and biomineralization of c-HAp and unc-HAp.

## Data Availability

The data presented in this study are available in article.

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
