# Peer review of "Comparative In Vitro Dissolution Assessment of Calcined and Uncalcined Hydroxyapatite Using Differences in Bioresorbability and Biomineralization"

_ijms, 2024, doi:10.3390/ijms25010621_

Round 1
Reviewer 1 Report
Comments and Suggestions for Authors
Review
Title: Comparison of Osteogenesis of Calcined and Uncalcined Hydroxyapatite using Differences in Bioresorbability and Biomineralization
Authors: Woo Young Jang, Jae Chul Pyun and Jeong Ho Chang
In this study, the authors report results regarding the osteogenesis comparison of calcined hydroxyapatite (c-HAp) and uncalcined hydroxyapatite (unc-HAp), evaluated by in vitro dissolution using the differences in bioresorption and biomineralization. Their findings revealed that Unc-HAp was characterized by higher particle size, surface area, and pore volume than c-HAp. The authors also investigated the osteogenesis of c-HAp and unc-HAp through their dissolution behavior in citric acid. Their results revealed that during dissolution, the weight of c-HAp and unc-HAp continued to increase, but the concentration of calcium ions released into solution increased and then decreased. Moreover, the calcium content and cell viability of unc-HAp were higher than c-HAp. Therefore, it can be said that the bioresorption and biomineralization of unc-HAp are higher than c-HAp, and therefore the osteogenesis effect is also higher.
The manuscript is well written and structured. In my opinion it could be considered for publication after the following improvements:
The introduction should be improved and the novelty of the present study should be emphasized compared to the existing ones.
Furthermore, the conclusions should be more concise and present the novelty of the findings.
In addition, in order to prove the osteogenesis, bioresorbability and biomineralization
of the materials, supplementary biological assays are needed. If the authors cannot provide them, please change the title of the manuscript as well the intended purpose of the study.
Regarding the biocompatibility of the samples, additional information should be added. Microscopic images of the cells after exposure to the samples are needed in order to see if the samples induce any morphological changes to the cells. More than that, if it is in the authors grasp, studies regarding the cellular uptake of nanoparticles should be presented.
Comments on the Quality of English LanguageMinor English improvements are welcomed.
Author Response
December 16, 2023
Dear Editorial Board of International Journal of Molecular Sciences:
:
A revised manuscript titled " Comparison of In vitro Dissolution Assessment of Calcined and Uncalcined Hydroxyapatite using Differences in Bioresorbability and Biomineralization" for publication in International Journal of Molecular Sciences.
Thanks very much for your favorable comments on our manuscript.
We have modified the manuscript accordingly, and the detailed corrections are listed in separate file point by point including the references and highlights. .
We are looking forward to your positive response for the review process.
Sincerely yours,
Jeong Ho Chang, Ph.D.
Center for Convergence Bioceramic Materials
Korea Institute of Ceramic Engineering and Technology
JHC:at
Encl/

Reviewer 2 Report
Comments and Suggestions for Authors
A comparison of the osteogenesis of calcined and uncalcined hydroxyapatite is being reported on here in a scientific study
As I read the paper, I did wonder what was original about it. This difference between calcined and uncalcined HAp is well known so what is the point of this study? I do not feel that the authors have properly scanned the literature to explore the previous work. The introduction gives no justification for the study.
Is this study filling a gap in the literature or is it a re invention of the wheel?
A thorough literature search is needed to 1) scan the literature to show where these differences in osteogenesis have come up in previous studies and 2) to show what the research gaps are that the authors' research is filling and to state that clearly in the aims.
In interpretations of their work they should be seeking to compare their work with other previously reported research that explores this theme to show what new information their studies have uncovered.
At the moment I do not see much originality here....
I am rejecting the paper in this form.
Comments on the Quality of English Language
English fine but the study is covering an unoriginal line of research. This difference is well known so why is this paper being written?
Author Response

(The authors gave the same response as above.)

Round 2
Reviewer 1 Report
Comments and Suggestions for Authors
The authors have responded to all my comments.
Comments on the Quality of English LanguageThe manuscript still need minor English adjustements.
Author Response
December 22, 2023
Dear Editorial Board of International Journal of Molecular Sciences:
:
A re-revised manuscript titled " Comparison of In vitro Dissolution Assessment of Calcined and Uncalcined Hydroxyapatite using Differences in Bioresorbability and Biomineralization" is enclosed with the hope that it will be considered for publication in International Journal of Molecular Sciences.
Thanks very much for your favorable comments on our manuscript.
We have modified the manuscript accordingly, and the detailed corrections are listed in separate file point by point including the references and highlights.
We are looking forward to your positive response for the review process.
Sincerely yours,
Jeong Ho Chang, Ph.D.
Center for Convergence Bioceramic Materials
Korea Institute of Ceramic Engineering and Technology
JHC:at
Encl/
Reviewer 2 Report
Comments and Suggestions for Authors
Thank you for the better explanation in the introduction but I think more work is needed to better explain this study. I have re evaluated it now and can see why you have done the study.
1) You wish to evaluate the difference in behaviour between unsintered and crystalline HAp
2) You have used a specific dissolution medium (because you are studying this in vitro) as a "tool" to help you understand the difference in behaviour.
3) This investigation has involved probing the "specific" dissolution chemistry of HAp in citric acid and studying in detail using various techniques. Based on this behaviour you have "extrapolated" this to a statement on the bioresorption and biomineralisation of HAp (in vitro and presumably in vivo??)
So I am still left confused by the following questions ....
1) How is the citric acid dissolution system relevant to in vivo processes (these are what counts in the end). Is how calcium citrate forms relevant to the actual bodily processes?
2) Why is citric acid used?
3) What about the real life processes?
The study is interesting and intriguing (now that I know its purpose) but I am wondering if there might be some (unintentional) disingenuousness in saying it explains the differences between unsintered and crystalline HAp? These are the results from a specific dissolution system and I did not see any caveat in the study which stated that this in vitro process was just a broad tool to help in the understanding and that the real life process could be very different
What about the role of carbonate? This is important and yet there is no mention. Carbonated HAp is more soluble than crystalline HAp.
Please can you respond to these point and provide a better explanation of the rationale of the study and how it might be "reflecting" the differences but not necessarily stating that this is what happens in vivo?
Other points:
The English is mostly good but causes some serious confusion in parts of the manuscript
The element "P" has three different spellings ! It is spelt PHOSPHORUS and not the other two spellings....it is also misspelt in the figure axis labels as well
There is no such word as "uncalcination" ....please correct
Some sentences are not expressed in a correct way which leads to confusion as to what is being stated in them.
At one point you stated that pore size "increases" for crystalline HAP?? This must be an error? When HAP is sintered its pore size should decrease surely?
See below :
"The surface 154 area and pore volume of c-HAp were much lower than those of unc-HAp, but its pore size was 155 higher. This phenomenon occurs as calcination progresses, leading to a denser internal 156 structure and the occlusion of some pores. Additionally, due to the influence of thermal energy, 157 particles were binded together, and in this process, the pore size can be further increased."
This sentence makes no sense:
"Therefore, both c-HAp and unc-HAp demonstrated favorable mechanical properties suitable 183 for use as biomaterials; so, they were performing in vitro dissolution assessments to confirmed 184 bioresorption and biomineralization"
who ae THEY?
Better justification of the study is needed and a caveat given as to comparability of the results obtained with real life
Comments on the Quality of English LanguageThe English is mostly good but causes some serious confusion in parts of the manuscript
The element "P" has three different spellings ! It is spelt PHOSPHORUS and not the other two spellings....it is also misspelt in the figure axis labels as well
There is no such word as "uncalcination" ....please correct
Some sentences are not expressed in a correct way which leads to confusion as to what is being stated in them.
At one point you stated that pore size "increases" for crystalline HAP?? This must be an error? When HAP is sintered its pore size should decrease surely?
See below :
"The surface 154 area and pore volume of c-HAp were much lower than those of unc-HAp, but its pore size was 155 higher. This phenomenon occurs as calcination progresses, leading to a denser internal 156 structure and the occlusion of some pores. Additionally, due to the influence of thermal energy, 157 particles were binded together, and in this process, the pore size can be further increased."
This sentence makes no sense:
"Therefore, both c-HAp and unc-HAp demonstrated favorable mechanical properties suitable 183 for use as biomaterials; so, they were performing in vitro dissolution assessments to confirmed 184 bioresorption and biomineralization"
who ae THEY?
Author Response

(The authors gave the same response as above.)

Round 3
Reviewer 2 Report
Comments and Suggestions for Authors
Minor changes to be attended to ............
Recently added explanations sound OK but a recent amendment needs improving upon because it is still vague in its expression and no one will understand properly what you mean especially when there seems to be a contradiction at the end....
"However, a typical in vitro dissolution assessment requires a long time to see the biomineralization process. In the in vivo assessment, osteoblasts play a role in facilitating this process, but not in the in vitro dissolution assessment. Therefore, the calcium ions released by in vitro dissolution with citric acid were easily biomineralized"
WHY does it require "a long time to see the biomineralisation process" and what is a "typical" in vitro process
"Therefore, the calcium ions released by in vitro dissolution with citric acid were easily biomineralized"
The above needs slight modification to emphasise that the SPECIFIC in vitro system you are studying will allow biomineralisation to occur at an accelerated so allowing its study for the materials in question.
Then it will be clear to the readers what you are doing in this research..
Figure 3 caption ...you refer to calcium and phosphorus IONS ....ICP-OES is an elemental analysis technique and does not detect "ions" ...the descriptions in Figure 3 about referring to "ions" are scientifically incorrect and must be changed. You should replace this with "calcium and phosphorus element concentrations" If the figure axis labels also contain the word "ions" these should also be removed
I should also add that there is no such thing chemically as "phosphorus ions"
Comments on the Quality of English Language
"Noncalcination" ......this is also not an appropriate Scientific term to use in the manuscript. This kind of statement is not used in normal practice to describe uncalcined hydroxyapatite. Its use in the manuscript is unnatural/unscientific English and should be replaced with something like
"the effect of not calcining hydroxyapatite" or "the effect of not subjecting hydroxyapatite to a calcination process"
This is better English and will be understood better by readers...
This recently added phrase :
"Therefore, c-HAp and unc-HAp were confirmed bioresorption and biomineralization through in vitro dissolution assessments. "
makes no sense at all.....what are you actually wanting to state here because what you wrote is not clear?
It needs to be changed.
